# How the water-soluble hemicarcerand incarcerates guests at room temperature decoded with modular simulations

Katherine G. McFerrin[1] & Yuan-Ping Pang [1✉]

Molecular dynamics simulations of hemicarcerands and related variants allow the study of constrictive binding and offer insight into the rules of molecular complexation, but are limited because three-dimensional models of hemicarcerands are tedious to build and their atomic charges are complicated to derive. There have been no molecular dynamics simulations of the reported water-soluble hemicarcerand (Octacid4) that explain how Octacid4 encapsulates guests at 298 K and keeps them encapsulated at 298 K in NMR experiments. Herein we report a modular approach to hemicarcerand simulations that simplifies the model building and charge derivation in a manner reminiscent of the approach to protein simulations with truncated amino acids as building blocks. We also report that in aqueous molecular dynamics simulations at 298 K *apo* Octacid4 adopts two clusters of conformations one of which has an equatorial portal open but the guest-bound Octacid4 adopts one cluster of conformations with all portals closed. These results explain how Octacid4 incarcerates guests at room temperature and suggest that the guest-induced host conformational change that impedes decomplexation is a previously unrecognized conformational characteristic that promotes strong molecular complexation.

[1] Computer-Aided Molecular Design Laboratory, Mayo Clinic, Rochester, MN, USA. ✉email: camdl1@icloud.com

Carcerands and their variants hemicarcerands are container molecules comprising two identical bowl-shaped resorcinarene fragments tethered rim-to-rim with four linkers (Fig. 1)[1–5]. These molecules, containing four equatorial portals in the linker region and two axial portals in the resorcinarene region, are developed as supramolecular hosts for potential synthetic molecular cells[6] or prototypic drug delivery systems[7,8]. Carcerands incarcerate their guests in the host cavity during the rim-to-rim tethering process in the host synthesis that entraps components of the reaction medium as permanent guests[1], but with hemicarcerands the guests enter and exit the host cavity at high temperatures and remain in the cavity at low temperatures[2–5]. The binding characteristics and solubility of (hemi)carcerands are governed mostly by the linker structures. The reported water-soluble hemicarcerand 4 (Octacid4; tethered with a 4,6-dimethylisophthalic acid fragment as shown in Fig. 1) is reportedly a functionally different hemicarcerand variant that can form complexes with small molecules in water without the need to adjust temperature[7,8]. Therefore, the Octacid4-like complexes are useful model systems for the study of constrictive binding, which is a type of molecular complexation that is affected by the activation energy required for a guest to pass through a narrow portal of the host cavity[3,9,10], to obtain insights into the rules of molecular complexation.

To date, only a few molecular dynamics (MD) simulations of hemicarcerands or their variants or complexes have been reported[11–14]. This scarcity is partly because hemicarcerands have more than 200 atoms, making their three-dimensional models tedious to build. The scarcity is also due to the complexity of hemicarcerands, all of which have multiple conformations[8,12,15]. As shown in Fig. 2, variations of the torsions of the four Octacid4 linkers can result in many distinct conformations, which complicates the derivation of the conformation-dependent atomic charges of Octacid4. These technical complexities, as detailed below, may explain why there have hitherto been no aqueous MD simulations of Octacid4 to understand how it unusually encapsulates various small-molecule guests at 298 K and keeps them encapsulated at the same temperature such that the bound guests

can be differentiated in their NMR spectra from those in the bulk phase[7].

Herein we report a modular approach to MD simulations of Octacid4 and its complexes that simplifies the model building and charge derivation. We also report the characterization of the host cavity and the guest motion inside the host in the MD simulations using the modular method. This characterization explains how Octacid4 incarcerates guests in a manner fundamentally different from those of carcerands and hemicarcerands, and offers mechanistic insight into molecular complexation.

## Results

**The modular approach.** In addition to the general technical complexities in hemicarcerand simulations noted above, the charge derivation for water-soluble hemicarcerands such as Octacid4 is particularly complicated. Because of the need to balance the atomic charges of the water-soluble hemicarcerand with the charges of the aqueous solvent and the charges of the small-molecule guest when using an AMBER forcefield such as FF12MClm[16], the hemicarcerand charges need to be derived from ab initio calculations using the HF/6-31G* basis set that uniformly overestimates the polarity of the molecule. This is because (1) aqueous solvent models (such as the widely used TIP3P empirical water model) include polarization effects due to the empirical calibration to reproduce the density and enthalpy of liquid vaporization[17], and (2) the small-molecule guest bears the restrained electrostatic potential (RESP) charges that are derived from ab initio calculations using the HF/6-31G* basis set[18–20]. Further, to obtain the atomic charges of a water-soluble hemicarcerand without any bias toward one particular conformation of the molecule, the hemicarcerand charges need to be obtained from ab initio calculations at the HF/6-31G*//HF/6-31G* level with multiple conformations of the molecule followed by using the Lagrange multiplier to force identical charges on equivalent atoms in these conformations[20]. These ab initio calculations and the Lagrange multiplier specification are computation demanding and labor intensive, respectively. For example, the ab initio calculation of one Octacid4 conformation at the HF/6-31G*//HF/6-31G* level took ~168 CPU hours using the computers at the University of Illinois Urbana-Champaign National Center for Supercomputing Applications.

During the course of our Octacid4 MD simulations, we recognized that the technical complexities described above differed little from those of MD simulations of proteins with large numbers of atoms and conformations. One known approach to circumventing the complexities of protein MD simulations is to perform the simulations using the modular concept underlying the second-generation AMBER forcefield[21]. This forcefield builds a protein model with truncated amino acids as building blocks and derives the atomic charges of each building block from its multiple conformations using the Lagrange multiplier to force identical charges on equivalent atoms in these conformations[20]. Given the effectiveness of the modular concept as demonstrated by the reported MD simulations for autonomous miniprotein folding that achieved agreements between simulated and experimental folding times within factors of 0.6–1.4[22], we used the modular concept to simplify the model building and charge derivation for Octacid4 and devised a modular method for the Octacid4 simulations. We describe below two key attributes of our method for performing the MD simulations of Octacid4.

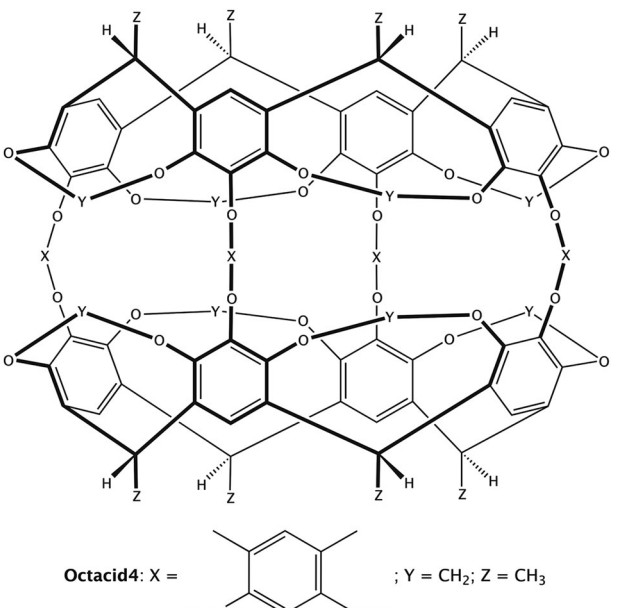

**Octacid4**: X =                 ; Y = CH₂; Z = CH₃

**Fig. 1 Chemical structures of hemicarcerands and Octacid4.**
Hemicarcerands carry a hydrophobic linker X, whereas Octacid4 has a 4,6-dimethylisophthalic acid fragment as a di-anionic linker that makes Octacid4 water soluble and functionally different from hemicarcerands.

**The building block of Octacid4.** Because Octacid4, similar to most hemicarcerands, has the $D_{4h}$ symmetry, we divided it into four identical building blocks (termed HC1 for the Octacid4 in the fully deprotonated form; Fig. 3). In this building block, C1

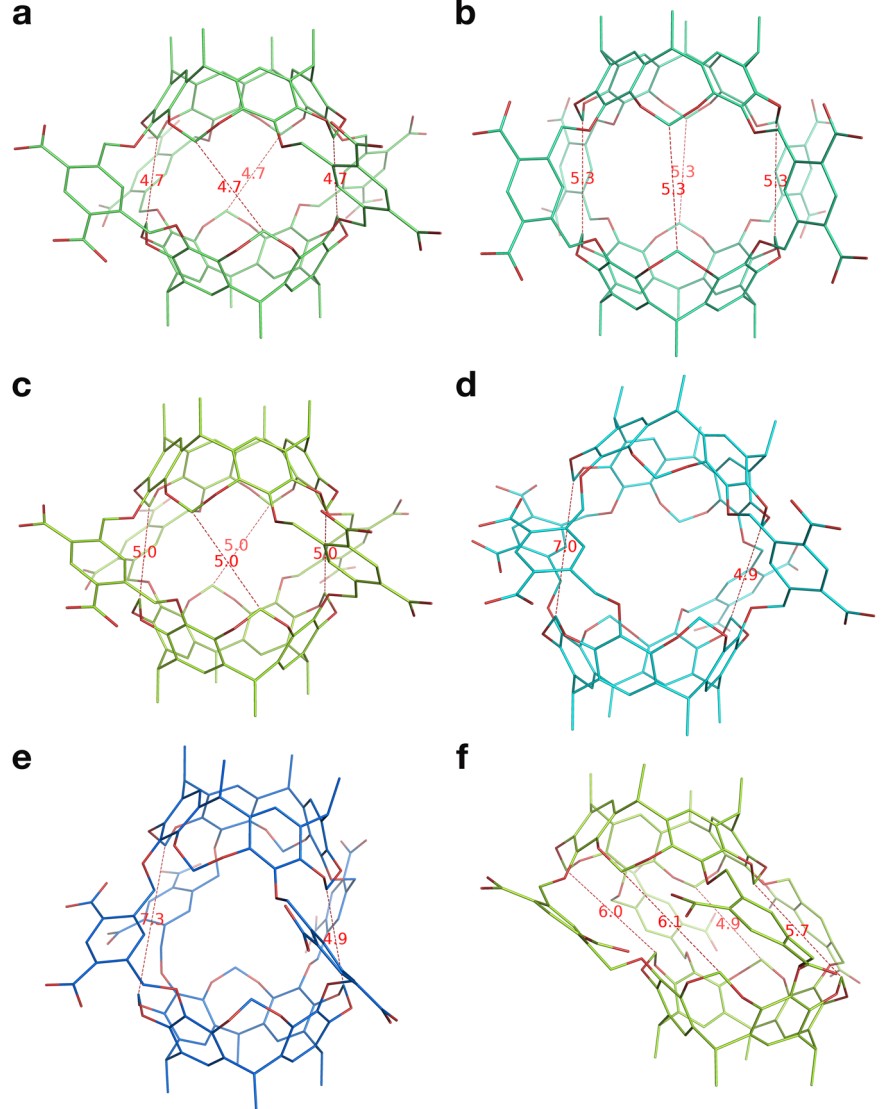

**Fig. 2 Distinct conformations of Octacid4.** Distances shown by thin dashed lines to indicate the open or closed equatorial portal are in Angstroms. Oxygen atoms are in red. Carbon atoms are in lime, green cyan, limon, cyan, marine, or yellow orange. Hydrogen atoms, counter ions and water molecules are not displayed for clarity. **a** Energy-minimization–derived Octacid4 in vacuo with all equatorial portals closed. **b** Energy-minimization–derived Octacid4 in vacuo with all equatorial portals open. **c** The most populated Octacid4 conformation with all equatorial portals closed in the aqueous MD simulations. **d** The second most populated Octacid4 conformation with one equatorial portal open in the aqueous MD simulations. **e** An Octacid4 conformation with one equatorial portal open in the aqueous MD simulations. **f** Another Octacid4 conformation with equatorial portals open in the aqueous MD simulations.

and C20 are designated as the head and tail atoms of the residue, respectively, similar to the designation of head and tail atoms in a truncated amino acid residue for protein simulations. Topology-wise, we built Octacid4 as shown in Fig. 3 with a sequence of HC1-HC1-HC1-HC1.

First, we assembled the four residues into a linear molecule. This assembly followed the same scheme as that for assembling protein residues—namely, constructing a covalent bond between the tail atom of the preceding residue and the head atom of the following residue. Second, we converted the linear molecule to a cyclic molecule by specifying a cross-link (viz., constructing a covalent bond) between the head atom of the first HC1 and the tail atom of the last HC1. Third, we converted the cyclic molecule to Octacid4 by specifying (i) four cross-links between C4 of the preceding HC1 and C18a of the following HC1, (ii) four cross-links between O5 of the preceding HC1 and C17a of the following HC1, and (iii) four cross-links between O16a of the preceding HC1 and C6b of the following HC1.

The Octacid4 sequence and all cross-links can be specified altogether, thus making the Octacid4 model building conceptually simple, algorithmic, and generalizable. Notably the HC1 building block can, in the same manner, be further divided into three sub-building blocks for hemicarcerands that use different linkers to tether the two resorcinarene fragments.

**Atomic charges of HC1.** Given the HC1 residue defined above, we obtained the atomic charges of Octacid4 via the RESP charge derivation for HC1 with the following specific conditions, in addition to the general conditions that are the same as those for protein RESP charges (such as the two-stage fitting for the methyl and methylene groups and forcing identical charges on equivalent intramolecular atoms)[20,21].

First, we converted HC1 to HCD1 by attaching blocking groups to the junction atoms of HC1 (Fig. 3) as these groups are needed to mimic the polar and aromatic groups abutting the junction atoms in Octacid4. We then performed the ab initio

**Fig. 3 Chemical structures of HC1 and HCD1 and the assembly of HC1 into the octa-anionic Octacid4.** Thick lines indicate the bonds between main-chain atoms. Thin lines indicate the bonds between side-chain atoms or between a main-chain atom and a side-chain atom. Thin dashed lines and colored lines indicate the respective intra- and inter-residue bonds that are constructed with cross-links. The double bonds and net charges are not displayed for clarity.

calculation of HCD1 to derive the HCD1 charges by using the Lagrange multiplier to force (1) HCD1 to have a net charge of −2 and (2) the total charge of all blocking groups to be zero, similar to the protein charge derivation from ab initio calculations of the acetyl- and N-methyl-blocked amino acids whose blocking groups are used to mimic adjacent residues of the central amino acid[20]. Second, to balance the Octacid4 charges with those of the

solvent and the guest, we obtained the HCD1 RESP charges from ab initio calculations of HCD1 at the HF/6-31G*//HF/6-31G* level. Third, to avoid any bias toward one particular conformation of Octacid4, we obtained the HCD1 RESP charges from ab initio calculations using (1) two HCD1 conformations taken from the representative conformations of Octacid4 in vacuo with all equatorial portals open (abbreviated as the open conformation;

**Table 1 Static and dynamic properties of Octacid4 and its complexes.**

| Structure | MDG (Å) | GMV (cm³/mol) | Temp (K) | Rg (Å) Mean ± SE | SAGS | MoGIO |
|---|---|---|---|---|---|---|
| Open *Apo* Octacid4 (vacuo) | – | – | – | 5.86 | – | – |
| Closed *Apo* Octacid4 (vacuo) | – | – | – | 5.37 | – | – |
| *Apo* Octacid4 (water) | – | – | 298 | 5.44 ± 0.02 | – | – |
| | | | 340 | 5.45 ± 0.02 | | |
| | | | 363 | 5.47 ± 0.02 | | |
| DMSO●Octacid4 (water) | 4.5 | 49 | 298 | 5.41 ± 0.02 | Ball | Free spin |
| | | | 340 | 5.41 ± 0.02 | | |
| | | | 363 | 5.43 ± 0.02 | | |
| 1,4-Dioxane●Octacid4 (water) | 4.8 | 74 | 298 | 5.37 ± 0.01 | Ball | Free spin |
| | | | 340 | 5.38 ± 0.01 | | |
| | | | 363 | 5.39 ± 0.01 | | |
| DMA●Octacid4 (water) | 5.3 | 73 | 298 | 5.40 ± 0.01 | Short rod | Partially free spin |
| | | | 340 | 5.41 ± 0.01 | | |
| | | | 363 | 5.42 ± 0.01 | | |
| EtOAc●Octacid4 (water) | 6.5 | 71 | 298 | 5.40 ± 0.01 | Short rod | Partially free spin |
| | | | 340 | 5.41 ± 0.01 | | |
| | | | 363 | 5.42 ± 0.01 | | |
| DEA●Octacid4 (water) | 6.8 | 72 | 298 | 5.64 ± 0.02 | Long rod | Axial spin |
| | | | 340 | 5.65 ± 0.02 | | |
| | | | 363 | 5.66 ± 0.02 | | |
| *p*-Xylene●Octacid4 (water) | 6.9 | 82 | 298 | 5.56 ± 0.02 | Long rod | Axial spin |
| | | | 340 | 5.59 ± 0.02 | | |
| | | | 363 | 5.60 ± 0.02 | | |
| Naphthalene●Octacid4 (water) | 7.1 | 115 | 298 | 5.65 ± 0.02 | Long rod | Axial spin |
| | | | 340 | 5.66 ± 0.02 | | |
| | | | 363 | 5.66 ± 0.02 | | |

*MDG* The maximal dimension of the guest calculated using Gaussian 98, *GMV* Guest molar volume calculated using Gaussian 98, *Temp* Temperature, *Rg* Radius of gyration obtained from 20 independent and distinct simulations, *SAGS* The shape of the average guest structure, *MoGIO* The motion of the guest incarcerated in Octacid4, *Axial spin* rotation around the axial axis, *Free spin* rotations evenly around multiple axes, *Partially free spin* rotations around one axis more frequently than around the orthogonal axis.

Fig. 2b) and all equatorial portals closed (abbreviated as the closed conformation; Fig. 2a) and (2) the Lagrange multiplier to force identical charges on equivalent atoms in the two conformations akin to the charge derivation method for proteins that uses the α-helical and β-strand side-chain conformations[20]. Last, we extracted the HC1 charges from the HCD1 charges.

In contrast to the computation-demanding and labor-intensive charge derivation for the intact Octacid4 molecule, the HC1 charge derivation can be readily performed. For example, the ab initio calculation of each of the two HCD1 conformations at the HF/6-31G*//HF/6-31G* level took 31–40 CPU hours, considerably less than the ~168 CPU hours for Octacid4 noted above.

**Characterization of the Octacid4 cavity.** Using the modular method described above, we performed 24 sets of 20 distinct, independent, unrestricted, unbiased, classical isobaric–isothermal, and 316-ns MD simulations of Octacid4 and its complexes with seven small-molecule guests at 298, 340, and 363 K. These simulations were performed with statistical relevance to investigate the change in the cavity volume of *apo* and guest-bound Octacid4 in water. A set of 20 simulations for each complex or Octacid4 had an aggregated simulation time of 6.32 µs. The seven guests were dimethyl sulfoxide (DMSO), ethyl acetate (EtOAc), dimethyl acetamide (DMA), 1,4-dioxane, diethylammonium (DEA), *p*-xylene, and naphthalene. These molecules were selected according to their calculated maximal dimensions and molar volumes (Table 1) from the 14 reported guests that formed complexes with Octacid4[7].

As apparent in Fig. 4, the Octacid4 cavity is confined by a set of atoms shown with the sphere model. Therefore, we used the variation in the radius of gyration (Rg) of these atoms to estimate the change in cavity volume. Overall, the standard errors for the average Rgs of the cavity for the 24 sets of MD simulations were either 0.01 or 0.02 Å (Table 1), demonstrating the convergence for each simulation of the 24 sets and the statistical rigor of these simulations. It is worth noting that we used 22 atoms of the HC1 residue rather than 88 atoms of the intact Octacid4 to define the Rg calculation, which shows the advantage of the modular method over the intact-molecule method.

For the simulations of the *apo* Octacid4 in water at 298, 340, and 363 K, according to their time series of Rg and average values described below, we found that (1) *apo* Octacid4 in water adopted two clusters of conformations, one similar to the closed conformation of *apo* Octacid4 in vacuo with a small cavity and the other similar to the open conformation of *apo* Octacid4 in vacuo with a large cavity, (2) the large-cavity conformations were much less populated in water than the small-cavity conformations, (3) the population of the large-cavity conformations in water increased as temperature increased, and (4) *apo* Octacid4 in water underwent cavity expansion and contraction at the three temperatures.

For the *apo* Octacid4 simulations at 298 K, the Rg of the cavity remained at ~5.4 Å and periodically spiked to >5.7 Å (Figs. 5 and S1), while the cavity Rgs of the open and closed conformations of Octacid4 in vacuo were 5.86 and 5.37 Å, respectively (Table 1). The average and standard error of the cavity Rg for the 20 simulations at 298 K were 5.44 and 0.02 Å, respectively (Table 1). For the simulations at 340/363 K, the Rg also remained at ~5.4 Å and spiked to >5.7 Å, but the frequency of the spikes was much higher at 340/363 K than that at 298 K (Figs. S1–S3),

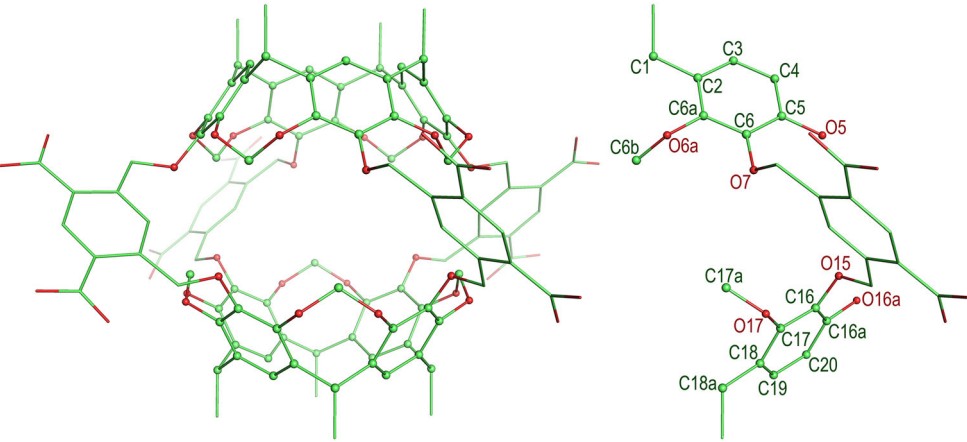

**Fig. 4 The atoms that confine the Octacid4 cavity.** All atoms that confine the cavity are shown with the sphere model. The names of the atoms in the HC1 residue that confine the cavity are C1, C2, C3, C4, C5, C6, C6a, O5, O6a, O7, C6b, C18a, C16, C17, C18, C19, C20, C16a, O15, O16a, O17, and C17a.

and the average and standard error of the Rg at 340/363 K were 5.45/5.47 and 0.02 Å, respectively (Table 1).

According to the conformational cluster analysis of the *apo* Octacid4 simulations in water at the three temperatures, we found that the most populated conformation (Figs. 2c, 6a, S4a, and S5a; population: 97.2% at 298 K, 97.8% at 340 K, and 95.1% at 363 K) in the MD simulations was nearly identical to the closed conformations in vacuo (Fig. 2a), whereas the second most populated conformation (population: 2.5% at 298 K, 2.1% at 340 K, and 4.7% at 363 K) was somewhat different from the open conformations in vacuo (Fig. 2b) in that the solution conformation had one equatorial portal open and the opposite portal closed (Fig. 2d).

For the simulations of the DMSO, EtOAc, DMA, and 1,4-dioxane complexes at 298, 340, and 363 K, according to their time series of Rg and average values described below, we found that (1) the complexes involving relatively compact guests in water all adopted one cluster of conformations that is similar to the closed conformation of *apo* Octacid4 in vacuo except for DMSO•Octacid4, which adopted two clusters conformations that resemble the open and closed conformations of *apo* Octacid4 in vacuo, but the population of the open conformation of DMSO•Octacid4 was much lower than that of *apo* Octacid4 in water, and (2) these complexes mostly kept their cavities contracted with all equatorial portals closed.

For the simulations with the compact guests, the cavity Rg of ~5.4 Å had a few or no spikes of >5.7 Å at the three temperatures (Figs. 5 and S1–S3). The respective average and standard error of the cavity Rg of each set of the 20 MD simulations at 298 K were 5.41 and 0.02 Å for DMSO, 5.40 and 0.01 Å for EtOAc, 5.40 and 0.01 Å for DMA, and 5.37 and 0.01 Å for 1,4-dioxane (Table 1). The corresponding values at 340/363 K were 5.41/5.43 and 0.02 Å for DMSO, 5.41/5.42 and 0.01 Å for EtOAc, 5.41/5.42 and 0.01 Å for DMA, and 5.38/5.39 and 0.01 Å for 1,4-dioxane (Table 1).

For the simulations of the DEA, *p*-xylene, and naphthalene complexes at 298, 340, and 363 K, according to their time series of Rg and average values described below, we found that (1) the complexes involving relatively bulky guests in water all adopted one cluster of conformations with their cavities larger than that of the closed conformation of *apo* Octacid4 in vacuo and smaller than that of the open conformation of *apo* Octacid4 in vacuo, and (2) these complexes mostly kept their cavities contracted with all equatorial portals closed.

For the simulations with the bulky guests, the cavity Rg of ~5.6 Å had a few or no spikes of >5.7 Å at the three

temperatures (Figs. 5 and S1–S3). The respective average and standard error of the cavity Rg of the simulations at 298 K were 5.64 and 0.02 Å for DEA, 5.56 and 0.02 Å for *p*-xylene, and 5.65 and 0.02 Å for naphthalene. The corresponding values at 340/363 K were 5.65/5.66 and 0.02 Å for DEA, 5.59/5.60 and 0.02 Å for *p*-xylene, and 5.66/5.66 and 0.02 Å for naphthalene (Table 1).

**Characterization of guest motion in Octacid4.** Overall, according to the conformational cluster analyses on the aqueous MD simulations for the seven Octacid4 complexes at 298, 340, and 363 K, as detailed below, we found that all seven guests spin— while keeping their terminal aliphatic/aromatic protons mostly close to the resorcinarene fragment—inside the Octacid4 cavity in all simulations. All complex simulations were performed using an energy-minimized initial conformation in which the host adopted the open conformation of *apo* Octacid4 in vacuo (Fig. 2b) and the guest was manually docked into the host cavity with its maximal dimension perpendicular to the axial axis (viz., the axis passing the two axial portals). In these initial conformations, the guests had an orientation with their terminal aliphatic/aromatic protons away from the resorcinarene fragment. However, during the simulations, all guests quickly adopted a new orientation with their maximal dimensions parallel to the axial axis thus making their terminal aliphatic/aromatic protons close to the resorcinarene fragment (Fig. 6). To confirm that the guests did not change from the new orientation back to the initial one, we extended our simulations from an aggregated simulation time of 6.32 to 31.6 μs for each of the four complexes with relatively compact and rigid guests (DMS, 1,4-dixoane, DMA, and *p*-xylene) by performing 100 316-ns MD simulations at 363 K for each complex, and found that none of these guests switched back to the initial orientation during the simulations. These simulation results are consistent with the noncovalent interaction gradient isosurfaces[23,24] shown in Fig. 7 that reveal more attractions between Octacid4 and the seven guests in the new orientation than in the initial orientation. The guest spin in the new orientation is interesting because, according to the Corey-Pauling-Koltun molecular model kit that was used in the reported NMR study[7], the new orientation prevents the guest from exiting the cavity due to the dimensions of the seven guests and the equatorial portal.

For the DMSO complex at the three temperatures, the most populated conformation had the guest oxygen atom pointing to the equatorial portals and the two guest methyl groups pointing to the axial portals (Figs. 6b, S4b, and S5b); in the average conformation of the largest conformation cluster, the guest was shrunk to a ball with the oxygen atom on one side and the two

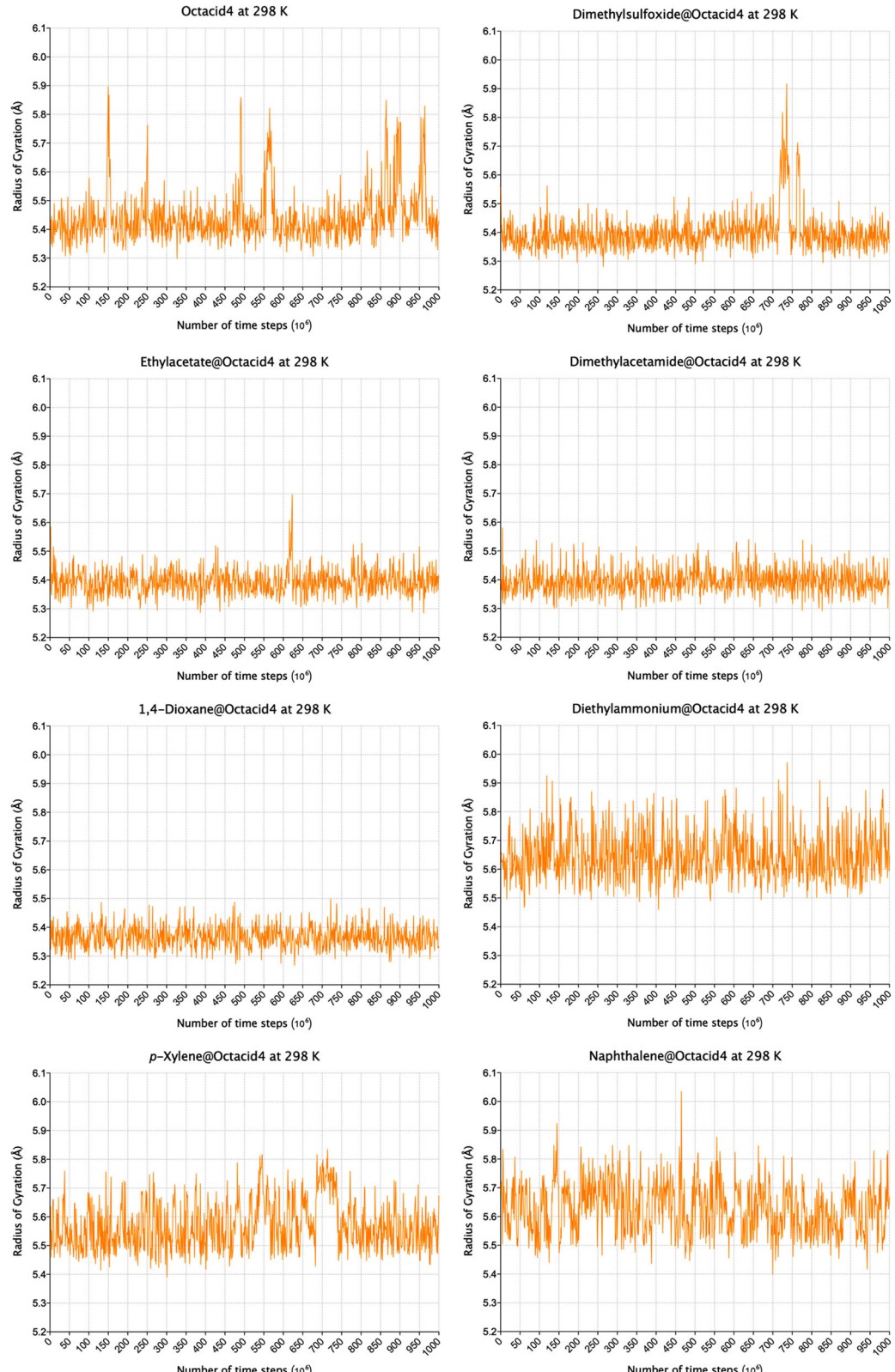

**Fig. 5 Time series of radius of gyration of the Octacid4 cavity.** Each time series was derived from the first of 20 distinct and independent MD simulations at 298 K (see Fig. S1 for those from all 20 simulations at 298/340/363 K).

overlapping carbon atoms on the other (Figs. 6b, S4b, and S5b), indicating that the guest had rotated around multiple axes.

For the EtOAc complex, the most populated guest conformation was partially extended at the three temperatures. The torsions of C1-C2-O1-C3 and C2-O1-C3-C4 as defined in Fig. 6c for this conformation were 102° and 89° at 298 K, −114° and 81° at 340 K, and −126° and 55° at 363 K, respectively. The population of this conformation was 99.9% at 298 K, 94.7% at

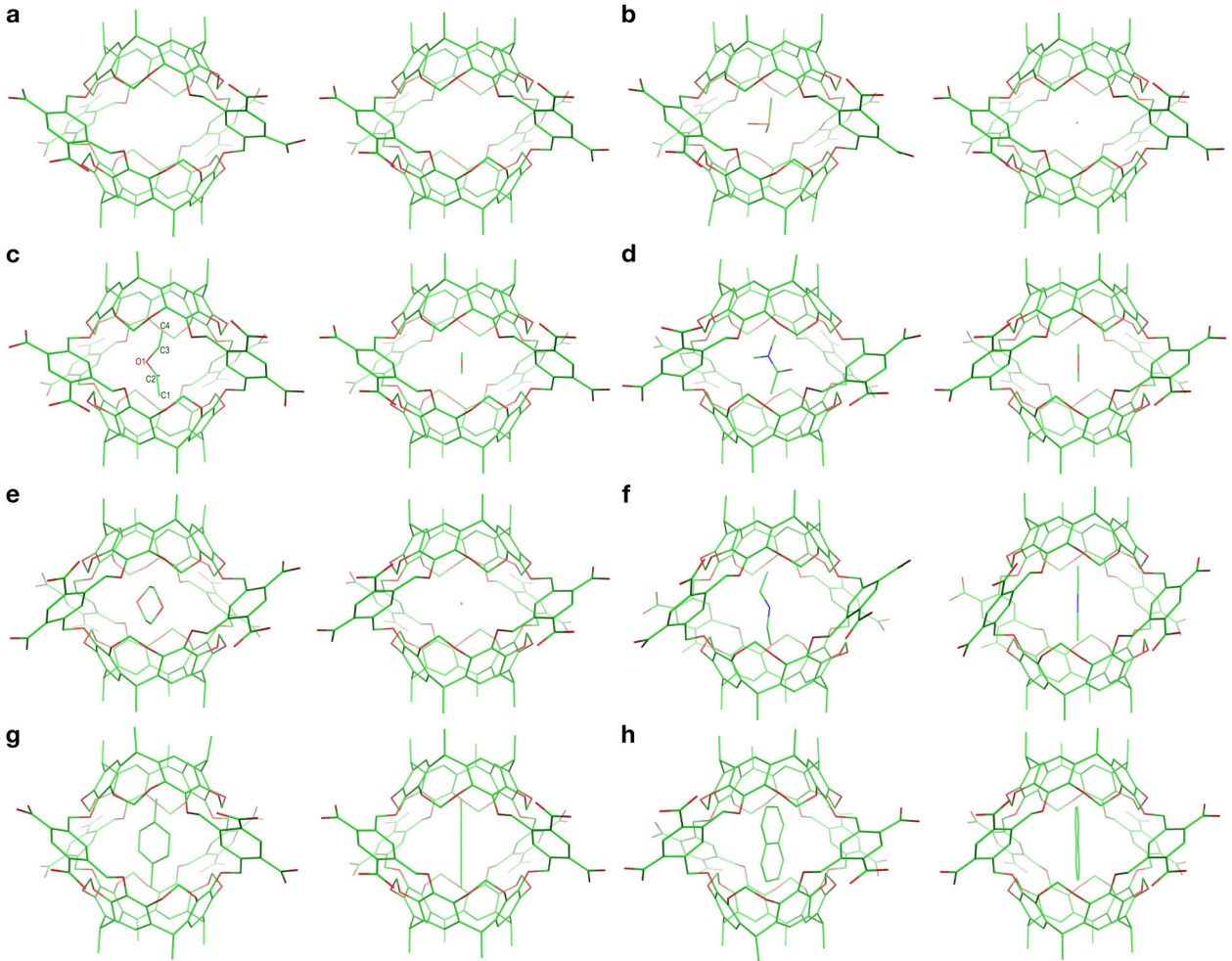

**Fig. 6 The most populated conformations of Octacid4 and its complexes in the MD simulations at 298 K. a** Octacid4. **b** DMSO•Octacid4.
**c** EtOAc•Octacid4. **d** DMA•Octacid4. **e** 1,4-Dioxane•Octacid4. **f** DEA•Octacid4. **g** *p*-Xylene•Octacid4. **h** Naphthalene•Octacid4. The representative
and average conformations in the largest conformation cluster of a set of 20 MD simulations for each complex are shown in the left and right panels,
respectively. No energy minimization was performed on these representative and average conformations. The sulfur, oxygen, nitrogen, and carbon atoms
are in yellow, red, blue, and green, respectively. Hydrogen atoms, counter ions and water molecules are not displayed for clarity.

340 K, and 99.4% at 363 K, respectively, whereas the population of the fully extended guest conformation with the corresponding torsions of 180 ± 30° was <0.1% at 298 K, 0.1% at 340 K, and 0.2% at 363 K. At the three temperatures, the most populated guest conformation had its two oxygen atoms pointing to the equatorial portals and its two methyl groups pointing to the axial portals (Figs. 6c, S4c, and S5c); in the average conformation of the largest conformation cluster, the guest was shrunk to a short rod with the oxygen atoms in the middle and the two carbon atoms on both ends of the rod (Figs. 6c, S4c, and S5c), indicating that the guest had rotated frequently around the axial axis and less frequently around the equatorial axis (viz., the axis passing the two opposing equatorial portals).

For the DMA complex at the three temperatures, the most populated conformation had the guest oxygen atom pointing to the equatorial portals and the two trans methyl groups of the guest pointing to the axial portals (Figs. 6d, S4d, and S5d); in the average conformation of the largest conformation cluster, the guest was shrunk to a short rod with (1) the oxygen atom on one side of the middle region, (2) the carbon atom trans to the oxygen atom on the other side of the middle region, and (3) the two trans carbon atoms on both ends of the rod (Figs. 6d, S4d, and S5d),

indicating that the guest had rotated frequently around the axial axis and less frequently around the equatorial axis.

At 298 and 340 K, the 1,4-dioxane complex had one conformation cluster. The guest in this cluster adopted the energetically stable chair conformation with the two guest oxygen atoms pointing to the equatorial portals and the two methylene groups on one side of the guest pointing to an axial portal while those on the opposite side pointing to the other axial portal (Figs. 6e and S4e), and in the average conformation of the cluster, the guest was shrunk to a ball with two oxygen atoms on opposite sides, indicating that the guest had rotated around multiple axes (Figs. 6e and S4e). At 363 K, the dioxane complex had two conformation clusters. The guest in the larger cluster (population: 75.0%) adopted the energetically less stable boat conformation (Fig. S5e), whereas the guest in the other cluster (population: 25.0%) had the energetically stable chair conformation. The guest in both clusters all had their two oxygen atoms pointing to the equatorial portals and the two methylene groups on one side of the guest pointing to an axial portal, while those on the opposite side pointing to the other axial portal (Fig. S5e). In the average conformations of the two clusters, the guests were all shrunk to a ball with two oxygen

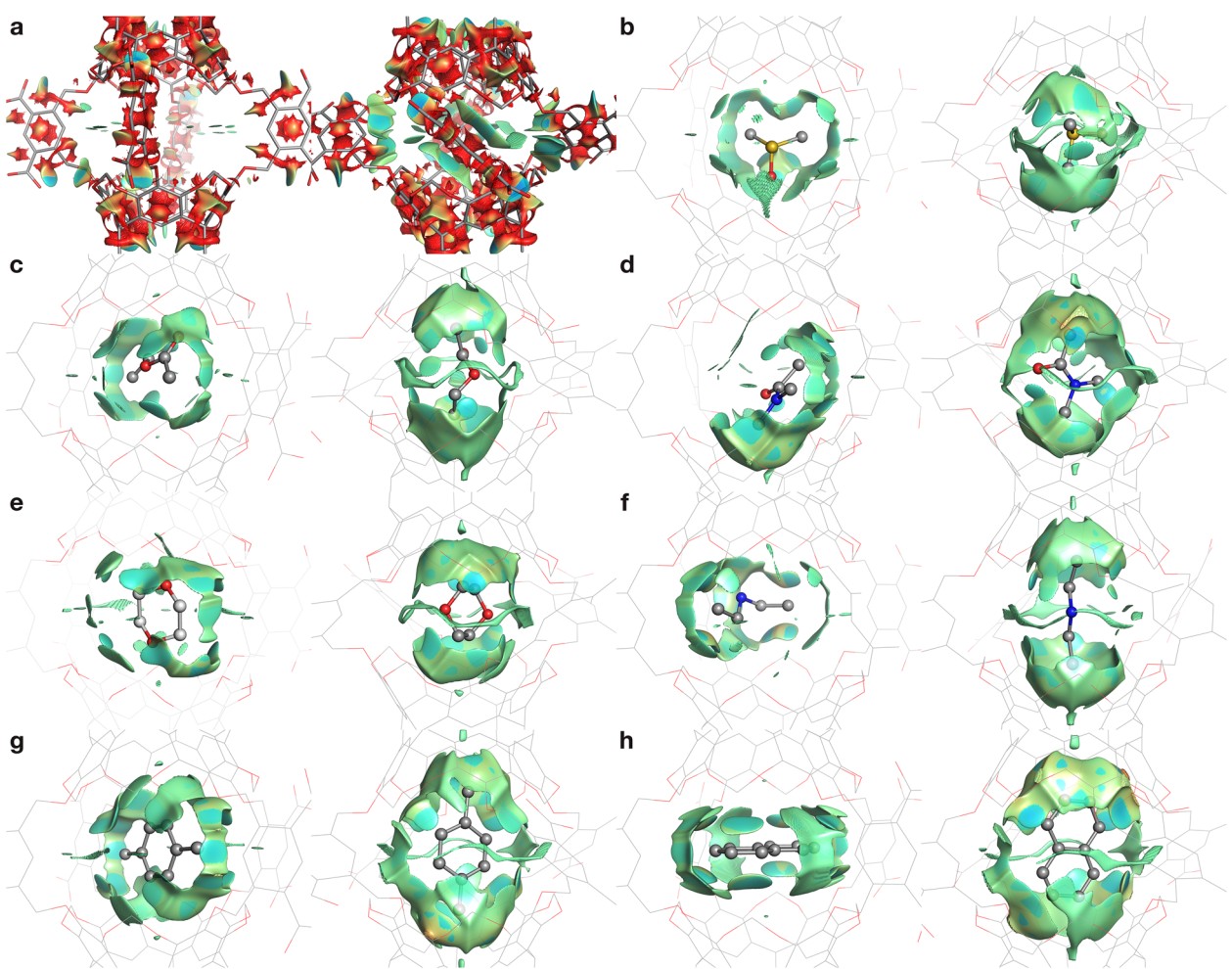

**Fig. 7 Noncovalent interaction gradient isosurfaces of Octacid4 and its complexes. a** Octacid4. **b** DMSO•Octacid4. **c** EtOAc•Octacid4.
**d** DMA•Octacid4. **e** 1,4-Dioxane•Octacid4. **f** DEA•Octacid4. **g** *p*-Xylene•Octacid4. **h** Naphthalene•Octacid4. The energy-minimized Octacid4
or its complex conformation as the initial conformation for a set of 20 MD simulations is shown on the left. The representative conformation in the largest
conformation cluster of the 20 MD simulations at 298 K is shown on the right. No energy minimization was performed on these representative
conformations. The sulfur, oxygen, nitrogen, and carbon atoms are in yellow, red, blue, and gray, respectively. Hydrogen atoms, counter ions, and water
molecules are not displayed for clarity. The gradient isosurfaces show the intramolecular (for **a**) and intermolecular (for **b**–**h**) interactions using a blue-red
scale with blue indicating strong attractions and red indicating strong repulsions.

atoms on opposite sides, indicating that the guest had rotated
around multiple axes (Fig. S5e).

The DEA, *p*-xylene, and naphthalene complexes had one
conformation cluster each at the three temperatures. The
conformations of these clusters had (1) the DEA nitrogen
atom pointing to the equatorial portals and the two DEA methyl
groups pointing to the axial portals, (2) the two *p*-xylene methyl
groups pointing to the axial portals, and (3) the naphthalene β-
carbon atoms pointing to the axial portals (Figs. 6f–h, S4f–h, and
S5f–h). In the average conformation of each cluster, the guest was
shrunk to a long rod whose length was the same as that of the
guest (Figs. 6f–h, S4f–h, and S5f–h), indicating that these guests
all had rotated exclusively around the axial axis.

## Discussion
**Support for the modular approach.** Our characterization studies
reveal that (1) *apo* Octacid4 adopted two clusters of conforma-
tions in aqueous MD simulations at 298 K, (2) the large cluster
had the conformations (population: 97.2%) nearly identical to the
one in vacuo with all equatorial portals closed, and (3) the small

one had the conformations (population: 2.5%) somewhat differ-
ent from the open conformation of *apo* Octacid4 in vacuo in that
the solution conformation had one equatorial portal open and the
opposite portal closed. The population ratio (97.2/2.5) of the two
clusters according to the conformational cluster analysis is con-
sistent with the frequency for the conformational exchanges
depicted in Fig. 5, which was derived independently from the Rg-
based analysis. In addition, all solution conformations derived
from the MD simulations have one resorcinarene fragment
slightly rotated with respect to the other to gain the attractions
between the aromatic linker and the resorcinarene fragment as
shown in Fig. 7a, which is consistent with reported crystal
structures of different complexes (involving the same or different
guests) all of which have one resorcinarene fragment rotated by
13–21° with respect to the other[2,9,25,26]. Moreover, the less
populated aqueous Octacid4 conformations with one equatorial
portal open and the opposite portal closed are also consistent with
the reported V-shaped hemicarcerand conformation proposed in
the sliding-door mechanism for complex formation[11,27]. These
internal and external consistencies indicate the plausibility of the
aqueous conformations of Octacid4 obtained a priori from our

modular-approach–based MD simulations and lend credence to our modular approach to Octacid4 simulations for probing solution conformations relevant to the following mechanistic insights.

**How Octacid4 incarcerates guests**. Our characterization studies also show that *apo* Octacid4 in water mostly adopted a cluster of conformations with small cavities whose equatorial portals were all closed and that, even at 298 K, this host also periodically and transiently adopted another cluster of conformations that had large cavities with at least one equatorial portal open. These periodic and transient conformations with an open equatorial portal explain how Octacid4 reportedly encapsulated guests in only a few minutes at 298 K in the NMR experiments[7], but do not explain how Octacid4 reportedly kept its guests encapsulated at the same temperature[7]. Perhaps the guests were incarcerated by strong intrinsic binding (viz., the complexation governed solely by intermolecular interactions between complexation partners as well as between each partner and solvent). However, the involvement of strong intrinsic binding is debatable given both the weak interaction of the cavity with water-soluble DMSO or water-soluble 1,4-dioxane as indicated by their free spins (Table 1 and Fig. 6b and e) and the noncovalent interaction gradient isosurfaces[23,24] shown in Fig. 7 that are mostly green rather than blue thus revealing weak intermolecular interactions between Octacid4 and its guests. An alternative explanation is therefore needed.

Unexpectedly, our characterization studies show that upon complexation Octacid4 adopted only one cluster of conformations. These conformations had, depending on the size of the guest, a contracted or slightly expanded cavity relative to that of the dominant conformation of *apo* Octacid4 in water. The conformations with contracted cavities had the equatorial portals all closed because the guests were relatively compact and subsequently the weak intermolecular interactions of the host cavity with a small guest shifted the host conformations with an open equatorial portal to the conformations with all equatorial portals closed. The conformations with slightly expanded cavities also had the equatorial portals all closed because the guests were relatively bulky and these guests jammed the cavity and consequently prevented the host from adopting the V-shaped sliding-door conformations shown in Fig. 2d, e to open any of the equatorial portals. Therefore, regardless of the size of the guest, the guest-bound Octacid4 in water adopted one cluster of conformations with all equatorial portals closed. These conformations satisfactorily explain how Octacid4 can incarcerate a range of guests at 298 K, which is the temperature at which the guests enter the cavity.

**Implications**. Collectively, our studies suggest that Octacid4 can open one of its equatorial portals without elevating temperature before complexation and close all of its equatorial portals without lowering temperature after complexation. These unexpected but interesting capabilities suggest that Octacid4 represents a class of supramolecular capsules that we call neocarcerands, which are capable of incarcerating guests in a manner fundamentally different from those of carcerands and hemicarcerands in that a neocarcerand can encapsulate its guest without the need to dissolve the guest in the reaction medium during the host synthesis or adjust temperature to open and close the host portal. These capabilities also suggest that the guest-induced host conformational change that impedes decomplexation through heightening the energy barrier for guest exit is a previously unrecognized

conformational characteristic that is conducive to strong molecular complexation. This characteristic could broaden the theoretical dissection of the experimentally observed complexation affinity and the design of new complex systems for materials technology, data storage and processing, catalysis, drug design and delivery, and medicine by accounting for not only intrinsic binding (which is limited because, confined by the time and cost of chemical synthesis[28], only a limited number of functional groups can be introduced to the guest to improve intermolecular interactions) but also constrictive binding (which requires one or a few functional groups to trigger the formation of a host conformation that hinders decomplexation as demonstrated by the Octacid4 complexes).

## Methods

**Molecular dynamics simulations**. Octacid4 or its complex neutralized with sodium ions was solvated with the TIP3P water[17] (solvatebox molecule TIP3BOX 8.2) using tLEaP of the AmberTools 16 package (University of California, San Francisco) and then energy-minimized for 100 cycles of steepest-descent minimization followed by 900 cycles of conjugate-gradient minimization to remove close van der Waals contacts using SANDER of the AMBER 11 package (University of California, San Francisco), forcefield FF12MClm[16], and a cutoff of 8.0 Å for noncovalent interactions. The tLEaP input file for building Octacid4 is provided in Supplementary Data 1. The resulting system was slowly heated to 298/340/363 K in 30 steps under constant temperature and constant volume, and then equilibrated for $10^6$ timesteps under a constant temperature of 298/340/363 K and the constant pressure of 1 atm employing isotropic molecule-based scaling. Finally, a set of 20 distinct, independent, unrestricted, unbiased, classical isobaric–isothermal, and 316-ns MD simulations of the equilibrated system was performed for the resulting Octacid4 or its complex using PMEMD of the AMBER 14/16/18 package (University of California, San Francisco), forcefield FF12MClm[16], and a periodic boundary condition at 1 atm and 298/340/363 K. The 20 unique seed numbers for initial velocities of the 20 simulations were taken from ref. [29]. All simulations used (i) a dielectric constant of 1.0, (ii) the Berendsen coupling algorithm[30] for thermostat and barostat, (iii) the particle mesh Ewald method to calculate electrostatic interactions of two atoms at a separation of >8 Å[31], (iv) $\Delta t = 1.00$ fs of the standard-mass time[16,32], (v) the SHAKE-bond-length constraint applied to all bonds involving hydrogen, (vi) a protocol to save the image closest to the middle of the "primary box" to the restart and trajectory files, (vii) a formatted restart file, (viii) the revised alkali ion parameters[33], (ix) a cutoff of 8.0 Å for noncovalent interactions, (x) a uniform 10-fold reduction in the atomic masses of the entire simulation system (both solute and solvent)[16,32], and (xi) default values of all other inputs of PMEMD.

The *apo* Octacid4 conformation with all equatorial portals open (Fig. 2b) was used as the initial conformation for the *apo* Octacid4 MD simulations. This conformation was obtained from energy minimization for 200 cycles of steepest-descent minimization followed by 2800 cycles of conjugate-gradient minimization using SANDER, forcefield FF12MClm[16], and a cutoff of 30.0 Å for noncovalent interactions. For each complex, the guest was manually docked into the cavity of the *apo* Octacid4 conformation with all equatorial portals open (Fig. 2b) in such a way that the maximal dimension of the guest was perpendicular to the axial axis; the resulting complex was then energy minimized using the same minimization protocol for obtaining the *apo* Octacid4 conformation; the energy-minimized complex was finally used as the initial conformation for the complex MD simulations.

FF12MClm was used in this study to investigate solution conformations of Octacid4 whose linkers can flip between left- and right-handed configurations and guest motions inside Octacid4. This was because of the effectiveness of FF12MClm in simulating the experimentally observed flipping between left- and right-handed configurations for C14–C38 of bovine pancreatic trypsin inhibitor in solution[16] and because of the need to compress the simulation time (viz., speed up simulations) by a factor of $10^{1/2}$ through 10-fold uniform reduction of the system mass. The hydrogen mass repartitioning scheme can also speed up simulations and was not used in this study because it affects dynamic properties of the system[34]. The forcefield parameters for the Octacid4 building block and for all small-molecule guests except dimethyl sulfoxide were obtained from ab initio calculations of the molecules at the HF/6-31G*//HF/6-31G* level using (1) a published procedure for the charge derivation[20] and (2) the arithmetic average from multiple conformations for the bond, angle, and torsion parameters. These parameters are provided in Supplementary Data 2. The forcefield parameters for dimethyl sulfoxide were obtained from ref. [35]. The forcefield parameters of FF12MClm are available in the Supporting Information of ref. [32].

All simulations were performed using a dedicated cluster of 100 12-core Apple Mac Pros with Intel Westmere (2.40/2.93 GHz) and computers at the University of Minnesota Supercomputing Institute and the Mayo Clinic high-performance

computing facility at the University of Illinois Urbana-Champaign National Center for Supercomputing Applications.

**ab Initio calculations**. All ab initio calculations were performed using Gaussian 98 (Revision A.7; Gaussian, Inc. Wallingford, CT), except those for the intact Octacid4 at the HF/6-31G*//HF/6-31G* level were done using Gaussian 16 (Revision C.01; Gaussian, Inc. Wallingford, CT).

**Conformational cluster analyses**. The conformational cluster analyses were performed using CPPTRAJ of the AmberTools 16 package (University of California, San Francisco) with the average-linkage algorithm[36], epsilon of 1.0 Å, and root mean square coordinate deviation on all carbon atoms of Octacid4 or its complex. Centering the coordinates of the complex and imaging the coordinates to the primary unit cell were performed prior to the cluster analyses. The Cartesian coordinates for representative and average conformations of Octacid4 and its complexes each of which was derived from the largest conformation cluster of a set of 20 316-ns MD simulations at 298/340/363 K as well as the corresponding initial conformations for the simulations are provided in Supplementary Data 3.

**Radius of gyration and noncovalent interaction gradient isosurface**. Radius of gyration was calculated using CPPTRAJ. All noncovalent interaction gradient isosurfaces were generated using the NCIPLOT program Version 4[23,24] with keywords of FINE and RANGE (3, −0.1 to −0.02, −0.02 to 0.02, and 0.02 to 0.1 au) for Fig. 7a or keywords of LIGAND (4.0 Å), FINE, and RANGE (3, −0.1 to −0.02, −0.02 to 0.02, and 0.02 to 0.1 au) for Fig. 7b–h.

## Data availability

Figs. S1–S5 and Supplementary Data 1–3 are provided in the Supplementary Information. Other relevant data are available from the corresponding author upon reasonable request.

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

## Acknowledgements

This work was supported by the US Army Research Office (W911NF-16-1-0264) and the Mayo Foundation for Medical Education and Research. Responsibility for the information and views in this study lies entirely with the authors. The authors acknowledge the computing resources provided by the University of Minnesota Supercomputing Institute and the Mayo Clinic high-performance computing facility at the University of Illinois Urbana-Champaign National Center for Supercomputing Applications.

## Author contributions

K.G.M. performed the literature search; prepared the z-matrices of the HC1 and HCD1 residues; performed energy minimization of HC1, HCD1, and Octacid4; analyzed the MD simulation result of p-xylene•Octacid4; contributed to revisions of the manuscript. Y.-P.P. conceived the modular method; designed the HC1 and HCD1 residues and all protocols for MD simulations and analyses; performed the remaining computational work; analyzed the data; wrote the manuscript.

## Competing interests

The authors declare no competing interests.
