## [Peer Review File · Communications Chemistry]

Reviewers' comments:

Reviewer #1 (Remarks to the Author):

This manuscript describes an interesting approach to the (hemi)carcerands as put forward by Cram in 1985 (JACS 1985, 107, 2575), and has developed force field parameters in a modular way. However, less emphasis is posed on how gating is involved (Science 1996, 273, 627), i.e. how the substrates enter and leave the inner cage of the supramolecule. Therefore, although the setup is well done, and is indeed a major advance for the study of these systems, the authors have not clearly shown whether the system deals with carcerands (substrates can enter, but can never leave) or hemicarcerands (substrates can be subtracted at elevated temperatures. From the results shown, the authors report on carcerands.

Hence, either the authors refer to the cages as carcerands, or they perform additional simulations to show how easy it is for the substrates to enter/leave the cage. For consistency, the latter approach would be preferred.

Minor points:

- p. 5, "we used to the modular concept" => remove "to"
- p. 12, in order to understand the binding of the substrates with the cage, it would be good to complement the study by a NCIplot study, which directly indicates H-bonding interactions, pi-pi stacking, and repulsion

Reviewer #2 (Remarks to the Author):

See attached file.

This is an interesting article showing, based on detailed molecular dynamics simulations, that hemicarcerand hosts adopt different conformations depending on the sizes and shapes of the guests they bind. This appears in fact to be the first report of simulations of these intriguing systems. The conformational results show encouraging agreement with experimental structure data. It also highlights some useful procedures for assigning force field parameters to these relatively large and complex host compounds. Several more detailed comments follow.

Especially given that this paper has a large methodological component, I think it would be enhanced if the Methods section briefly stated the rationale for using the FF12MC1M force field. What makes this particularly suitable? Also, again a sentence or two explaining the reduced mass concept would be helpful; I found it in the author's prior publications, but this paper could easily fit a brief explanation.

The Method section says that "Berendsen coupling" was used. Does this refer to the thermostat, the barostat, or both? Although use of the Berendsen methods need not be disqualifying, the authors may want to reconsider this selection for future simulations, as the Berendsen thermostat and barostat are known to give somewhat incorrect ensembles. See, e.g.

https://en.wikipedia.org/wiki/Berendsen_thermostat,
<http://www.pages.drexel.edu/~cfa22/msim/node33.html>.

A few paragraphs, notably those from "For the simulations of the apo Octacid4..." (page 7) to "with all equatorial portals closed" (page 9) start off with dry recitations of numbers that are hard to absorb and that are introduced without obvious motivation. In each paragraph, the point of the recitation becomes clear later in the paragraph, with sentences in every case starting "These time series of Rg...". Just as a matter of style, you might consider moving the key points to the start of each paragraph, and then using the recitation of details (or even better, pointers to these data in tables and figures) to back them up. This will help the reader remain motivated and oriented.

As a matter of scientific speculation... do the authors think that the cavity is large in vacuo due to the unshielded coulombic repulsions among the carboxylates? In water, these interactions would be greatly weakened because of water's high dielectric constant, perhaps explaining why the cavity tends to close more in water. Even further, at higher temperature, there may be two factors explaining the more open conformations. First, the greater kinetic energy may simply cause the molecules to explore some conformations that are too unstable to be visited at low temperature. Second, the dielectric constant of water falls with increasing temperature, and this might cause the carboxylate repulsions to become stronger, somewhat as in vacuo, again tending to push the system open.

Finally, I searched the SI in vain for files with representative 3D structures of the molecules and topology files with the force field parameters. It would be great to include these. Or perhaps they are present, and I missed them, in which case perhaps they could be more clearly identified.

--Mike Gilson

Response to the reviewers' comments

We greatly appreciate both reviewers' precious time and their insightful and stimulating comments. Below is our point-to-point response to all comments from the two reviewers. In this submission changes made according to the comments from Reviewers #1 and #2 are colored in red and magenta, respectively.

Reviewer #1 (Remarks to the Author):

This manuscript describes an interesting approach to the (hemi)carcerands as put forward by Cram in 1985 (JACS 1985, 107, 2575), and has developed force field parameters in a modular way. However, less emphasis is posed on how gating is involved (Science 1996, 273, 627), i.e. how the substrates enter and leave the inner cage of the supramolecule. Therefore, although the setup is well done, and is indeed a major advance for the study of these systems, the authors have not clearly shown whether the system deals with carcerands (substrates can enter, but can never leave) or hemicarcerands (substrates can be subtracted at elevated temperatures. From the results shown, the authors report on carcerands. Hence, either the authors refer to the cages as carcerands, or they perform additional simulations to show how easy it is for the substrates to enter/leave the cage. For consistency, the latter approach would be preferred.

Response: We have now performed eight sets of 20 316-ns MD simulations (with an aggregated simulation time of 6.32 μ s) for the *apo* Octacid₄ and each of all seven guests in complex with Octacid₄ at 363 K which is close to the boiling point of water. These simulations have confirmed our observation from previous simulations that all seven guests did not leave the cage in the MD simulations. We have also performed four sets of 100 316-ns MD simulations (with an aggregated simulation time of 31.6 μ s for each set) for four relatively compact and rigid guests (dimethyl sulfoxide, dimethyl acetamide, 1,4-dioxane, and *p*-xylene) in complex with Octacid₄ at 363 K. In these simulations, the guests did not leave the cage, nor did they change to the equatorial guest orientation (with the guest's greatest dimension perpendicular to the axial axis of the host). According to the CPK molecular model kit that was used in the reported NMR study, the equatorial guest orientation is required for the exit of the guest due to the dimensions of the guest and the equatorial portal. We have incorporated the new simulations results in the revision.

However, regardless whether the guests leave the cage at elevated temperatures, Octacid₄ encapsulates a guest without the need to dissolve the guest in the reaction medium for the host synthesis. For this reason Octacid₄ is not a carcerand because by definition a carcerand incarcerates a guest during the host synthesis that entraps a component of the reaction medium as a permanent guest (JACS 107, 2575, 1985). According to two independent experimental studies showing that unlike hemicarcerands Octacid₄ can incarcerate guests without adjusting temperature (Chem Commun 497, 1997; JACS 132,16423, 2010) and our manuscript explaining how Octacid₄ can incarcerate guests without adjusting temperature, Octacid₄ is not a hemicarcerand either.

Therefore, in view of this reviewer's insightful comment above, in the last paragraph of Discussion, we suggested the term "neocarcerands" for Octacid₄ and the likes in order to distinguish (hemi)carcerands from an interesting class of Octacid₄-like supramolecular capsules that are capable of strong host-guest complexation enabled by the guest-induced host

conformational change that impedes decomplexation. We have accordingly made changes of our manuscript including the title.

Minor points:

- p. 5, "we used to the modular concept" => remove "to"

- p. 12, in order to understand the binding of the substrates with the cage, it would be good to complement the study by a NCIplot study, which directly indicates H-bonding interactions, π - π stacking, and repulsion

Response: We have deleted "to" and added Fig. 7 and the description of our NCILOT study.

Reviewer #2 (Remarks to the Author):

This is an interesting article showing, based on detailed molecular dynamics simulations, that hemicarcerand hosts adopt different conformations depending on the sizes and shapes of the guests they bind. This appears in fact to be the first report of simulations of these intriguing systems. The conformational results show encouraging agreement with experimental structure data. It also highlights some useful procedures for assigning force field parameters to these relatively large and complex host compounds. Several more detailed comments follow.

Especially given that this paper has a large methodological component, I think it would be enhanced if the Methods section briefly stated the rationale for using the FF12MC1M force field. What makes this particularly suitable? Also, again a sentence or two explaining the reduced mass concept would be helpful; I found it in the author's prior publications, but this paper could easily fit a brief explanation.

Response: In the revised Methods section we have added the following explanation:

"FF12MC1m was used in this study to characterize solution conformations of Octacid₄ whose linkers can flip between left- and right-handed configurations and guest motions inside Octacid₄. This was because of the effectiveness of FF12MC1m in simulating the experimentally observed flipping between the left- and right-handed configurations for C₁₄-C₃₈ of bovine pancreatic trypsin inhibitor in solution (Proteins 84, 1490, 2016) and because of the need to compress the simulation time (*viz.*, speed up simulations for this computationally intensive study) by a factor of 10^{1/2} through 10-fold uniform reduction of the system mass. The hydrogen mass repartitioning scheme can also speed up simulations but was not used in this study because it affects dynamic properties of the system (J. Comput. Chem. 20, 786,1999)."

The Method section says that "Berendsen coupling" was used. Does this refer to the thermostat, the barostat, or both? Although use of the Berendsen methods need not be disqualifying, the authors may want to reconsider this selection for future simulations, as the Berendsen thermostat and barostat are known to give somewhat incorrect ensembles. See, e.g. https://en.wikipedia.org/wiki/Berendsen_thermostat, <http://www.pages.drexel.edu/~cf422/msim/node33.html>.

Response: We have clarified in the revision that the Berendsen coupling algorithm was for both. The issue the reviewer referred to on the Berendsen barostat is noted in the AMBER manual. This was why we used multiple steps to slowly heat the system to target temperature using constant volume. For Octacid₄-*p*-xylene, we actually simulated the system using both Berendsen and Langevin thermostats and found no difference between the two. Therefore we used the Berendsen thermostat in this project. However, as suggested by this reviewer, for simulations of large proteins, we mostly use the Langevin thermostat.

A few paragraphs, notably those from “For the simulations of the apo Octacid₄... “ (page 7) to “with all equatorial portals closed” (page 9) start off with dry recitations of numbers that are hard to absorb and that are introduced without obvious motivation. In each paragraph, the point of the recitation becomes clear later in the paragraph, with sentences in every case starting “These time series of Rg...”. Just as a matter of style, you might consider moving the key points to the start of each paragraph, and then using the recitation of details (or even better, pointers to these data in tables and figures) to back them up. This will help the reader remain motivated and oriented.

Response: We have moved the key points to the beginning of these paragraphs.

As a matter of scientific speculation... do the authors think that the cavity is large in vacuo due to the unshielded coulombic repulsions among the carboxylates? In water, these interactions would be greatly weakened because of water’s high dielectric constant, perhaps explaining why the cavity tends to close more in water. Even further, at higher temperature, there may be two factors explaining the more open conformations. First, the greater kinetic energy may simply cause the molecules to explore some conformations that are too unstable to be visited at low temperature. Second, the dielectric constant of water falls with increasing temperature, and this might cause the carboxylate repulsions to become stronger, somewhat as in vacuo, again tending to push the system open.

Response: This is a stimulating question. The carboxylates are at least 9 Å apart, but there are eight carboxylates in Octacid₄. With the bidentate chelation to a counterion, two carboxylates can be ~3 Å apart, but then the repulsion between the two is balanced by the attraction between the carboxylate and the counterion. An NMR binding experiment using different ionic strengths of a deuterated buffer may offer a better answer.

Finally, I searched the SI in vain for files with representative 3D structures of the molecules and topology files with the force field parameters. It would be great to include these. Or perhaps they are present, and I missed them, in which case perhaps they could be more clearly identified.

Response: In the SI of this submission, we have provided (1) the Cartesian coordinates for both representative and average 3D structures derived from the MD simulations as well as the initial 3D structures for the MD simulations, (2) the tLEaP input file for building Octacid₄ using the AMBER HC₁ library and frcmod files, and (3) the AMBER library and frcmod files for HC₁, EtOAc, DMA, 1,4-dioxane, DEA, *p*-xylene, and naphthalene.

REVIEWERS' COMMENTS:

Reviewer #1 (Remarks to the Author):

The authors have done a satisfactory revision, taking into account the comments raised, and therefore I can now recommend publication as is.

Reviewer #2 (Remarks to the Author):

I feel the authors have addressed the concerns raised regarding the initial submission.

TWO REVIEWERS' COMMENTS ON THE REVISION

Reviewer #1 (Remarks to the Author):

The authors have done a satisfactory revision, taking into account the comments raised, and therefore I can now recommend publication as is.

Reviewer #2 (Remarks to the Author):

I feel the authors have addressed the concerns raised regarding the initial submission.

RESPONSE TO THE REVIEWERS' COMMENTS ON THE REVISION

We greatly appreciate both reviewers' precious time and their positive comments.